# BaNEL: Exploration Posteriors for Generative Modeling using Only Negative Rewards

## Abstract

Today's generative models thrive with large amounts of supervised data and informative reward functions characterizing the quality of the generation. They work under the assumptions that the supervised data provides knowledge to pre-train the model, and the reward function provides dense information about how to further improve the generation quality and correctness. However, in the hardest instances of important problems, two problems arise: (1) the base generative model attains a near-zero reward signal, and (2) calls to the reward oracle are expensive. This setting poses a fundamentally different learning challenge than standard reward-based post-training. To address this, we propose BaNEL (Bayesian Negative Evidence Learning), an algorithm that post-trains the model using failed attempts only, while minimizing the number of reward evaluations (NREs). Our method is based on the idea that the problem of learning regularities underlying failures can be cast as another, in-loop generative modeling problem. We then leverage this model to assess whether new data resembles previously seen failures and steer the generation away from them. We show that BaNEL can improve model performance without observing a single successful sample on several sparse-reward tasks, outperforming existing novelty-bonus approaches in success rate, while using fewer reward evaluations.

## 1 Introduction

Today's generative models thrive with large amounts of supervised data and informative reward functions characterizing the quality of the generation, especially for generating language, image, video, and audio. This pipeline works well under the assumptions that 1) the supervised data provides broad enough coverage of the generation space, and 2) the reward function provides information about how to improve or focus the generation quality and correctness. Language modeling with verifiable rewards (Guo et al., 2025) works well because the base models often start with at least some positive reward signal on the task.

**Challenge: Tasks with near-zero reward and expensive reward oracles.** In many unsolved critical applications—including the next generation of theorem proving, algorithmic problem solving, and drug discovery, to name a few—this standard pipeline encounters two core challenges. (1) *Sparsity:* Oftentimes, the base generative model attains a near-zero reward signal. The probability of producing a positive-reward sample can be so low that the model may go through most of training without ever encountering one. (2) *High-cost reward evaluation:* Calls to the reward oracle can be expensive or risky, requiring costly simulations, computations, or even physical experiments (Korshunova et al., 2022). Hence, there is a need for **learning algorithms that can learn from exclusively negative-reward samples, while minimizing number of reward evaluations (NREs).** This setting poses a fundamentally different learning challenge than standard reward-based post-training. Learning in such harsh conditions is crucial: failure to tackle this challenge would mean that post-training is merely limited to distribution sharpening rather than unlocking genuinely new capabilities.

The performance of such learning algorithms largely depends on their ability to recognize and generalize from a small number of failures; ideally, this ability should **scale with compute.** In deep RL, reward sparsity is often addressed by introducing novelty bonuses to encourage exploration. Two of the most popular techniques for doing so include count-based methods (Bellemare et al., 2016; Ostrovski et al., 2017) and random network distillation (Burda et al., 2019). These methods have

proven effective in sparse-reward Atari environments such as Montezuma's Revenge (Ostrovski et al., 2017; Burda et al., 2019; Badia et al., 2020b;a). However, quality of the intrinsic signal does not scale with compute, and as such they must query the reward oracle frequently. On the other hand, prediction-error approaches (Schmidhuber, 2010; Pathak et al., 2017; Stadie et al., 2015) learn the dynamics of the environment; these methods can be scalable but they are inapplicable for training generative models, where the transition dynamics are known and deterministic. Recent reward-based sampling methods like GFlowNets (Bengio et al., 2021) allow for multiple parameter updates per reward evaluation, but they are unable to learn in extremely sparse environments.

**Our approach: Train a generative model on failures and update the policy distribution away from the negative samples.** The zero-reward problem can be solved in many ways, such as using positive transfer from other tasks or domains, hand-designing curricula, and/or engineering more informative and dense reward functions. We argue there will always fundamentally be tasks and settings where the base model attains an extremely sparse reward, and that even these negative samples provide useful information to learn and explore from. Motivated by other sparse reward reinforcement learning methods, we propose to use the negative samples and reweight the base distribution away from them. Specifically, we train a generative model on negative samples for multiple epochs, and use it to assess whether data is similar to previously seen failures. If a sample is similar to other zero-reward data, the algorithm rejects it before querying the expensive reward oracle. This mirrors human scientists who, based on their failures, know what is unlikely to work and thus what to try next.

In summary, we make the following contributions:

1. **Conceptual:** We show in Section 3 why existing leading techniques for post-training generative models and learning under sparse rewards do not apply to our extremely sparse, black-box setting, where calls to the reward oracle are costly.

2. **Algorithmic:** We present BaNEL (Bayesian Negative Evidence Learning), which offers three fundamental advantages for learning in extreme sparsity while minimizing calls to the reward oracle (Section 4). First, unlike other sparse-RL methods, it allows multiple parameter updates per each collected experience, allowing the model to learn efficiently from a handful of failures. Second, it provides a sequential exploration framework that systematically narrows the search space until finding initial successes. Third, unlike many sparse RL methods, BaNEL is based on Bayesian updates which modify the prior multiplicatively and never explicitly decrease the model's likelihood for failed attempts, better preserving the model's pre-trained knowledge.

3. **Evaluation:** We propose new experimental settings that enable controlled testing of exploration strategies for post-training generative models under sparse-reward conditions. We evaluate BaNEL in these sparse environments and tasks in Section 5. Our experiments suggest that BaNEL achieves a success rate on challenging problems higher than existing baselines for the same NRE budget; moreover, it enables trading off computation for success rate, in a new form of compute scaling.

## 2 PROBLEM FORMULATION: EFFICIENT LEARNING FROM SPARSE REWARDS

Let $\mathcal{V}$ be the discrete token set and $\mathcal{V}^*$ be the set of all finite strings over $\mathcal{V}$. Define the probability distribution of our **pre-trained generative model** as $p_{\boldsymbol{\theta}} : \mathcal{V}^* \to [0, 1]$ satisfying $\sum_{\mathbf{x} \in V^*} p_{\boldsymbol{\theta}}(\mathbf{x}) = 1$ with parameter $\boldsymbol{\theta}$. We further assume a given, binary **reward function** $r : \mathcal{V}^* \to \{0, 1\}$, where 1 and 0 mean success and failure, respectively. The success rate of the model $\rho(p_{\boldsymbol{\theta}})$ is defined as $\rho(p_{\boldsymbol{\theta}}) := \sum_{\mathbf{x}} p_{\boldsymbol{\theta}}(\mathbf{x}) r(\mathbf{x})$.

The goal of reward-based training is to further improve $\rho(p_{\boldsymbol{\theta}})$ without any additional supervised data. In particular, we assume that evaluating $r$ is costly or risky—for instance, this can occur when running clinical trials in drug development, performing large-scale simulations (Korshunova et al., 2022), or other cases involving direct interaction with the real world.

**Problem Statement.** Consider a pre-trained $p_{\boldsymbol{\theta}}$ with a success rate $\rho(p_{\boldsymbol{\theta}})$ that is so low that the model *does not encounter positive examples* during training with high probability. Our goal is to find

| Method | | Functionality | Low NREs |
|---|---|:---:|:---:|
| Policy Gradient | Classic | ○ | ○ |
| | Negative RL | ○ | ○ |
| Intrinsic Rewards | RND | ◐ | ○ |
| | Count-based methods | ◐ | ○ |
| GFlowNets | | ○ | ● |
| BaNEL (Ours) | | ● | ● |

Table 1: Comparison of desired properties from Section 3—functionality and low number of reward evaluations (NREs)—for key categories of learning methods. An empty circle ○ means the property is not satisfied, a filled circle ● means satisfied, and a half-filled circle ◐ means partially satisfied (e.g., a method is functional, but success rate does not increase much).

a new model $p_{\boldsymbol{\eta}}$ parameterized by $\boldsymbol{\eta}$ such that success rate $\rho(p_{\boldsymbol{\eta}}) \gg \rho(p_{\boldsymbol{\theta}})$, while minimizing the number of calls to the reward oracle $r$, which we denote number of reward evaluations (NREs).

Note that we are *not* necessarily trying to minimize overall computation—we want to minimize NREs, but we are willing to scale (increase) compute to make better use of reward-labeled samples.

## 3 EXISTING METHODS FAIL TO ADDRESS EXTREME REWARD SPARSITY

Our problem formulation requires algorithms to satisfy two properties:

1. **Functionality:** Does the algorithm improve upon the prior success rate in the extremely sparse setting, i.e., does the algorithm result in $\rho(p_{\boldsymbol{\eta}}) \gg \rho(p_{\boldsymbol{\theta}})$, given enough calls to the reward oracle?

2. **Low number of reward evaluations (NRE):** Does the algorithm make efficient use of the reward oracle $r$, e.g., by conducting multiple iterations of learning per reward evaluation?

We consider several categories of algorithms with respect to our problem requirements. Our high-level assessment of these methods is included in Table 1, with a more in-depth explanation below. Additional related work can be found in Appendix A.

### 3.1 WARM-UP EXAMPLE: POLICY GRADIENT

We start with the well-known policy gradient (Williams, 1992), the most common approach for post-training generative models from reward functions. It has achieved great success in challenging real-world tasks, including code synthesis and math problem solving (Guo et al., 2025).

**Classic policy gradient: zero rewards produce zero gradient** Under classic policy gradient, we draw $m$ samples $(\mathbf{x}_1, \ldots, \mathbf{x}_m)$, where $\mathbf{x}_i \sim p_{\boldsymbol{\theta}}$. If all of these samples receive zero reward, the standard REINFORCE policy gradient is zero: $\frac{1}{m} \sum_{i=1}^{m} r(\mathbf{x}_i) \nabla_{\boldsymbol{\theta}} \log p_{\boldsymbol{\theta}}(\mathbf{x}) = 0$. In this setting, policy gradient becomes brute-force random sampling until discovering the first rare success. By definition, this cannot improve success rate over $\rho(p_{\boldsymbol{\theta}})$. Moreover, we cannot update our model more than once per reward evaluation without resorting to other off-policy learning techniques.

**Negative RL** A straightforward way to enable learning is to subtract a constant baseline of 1:

$$\sum_{i=1}^{m} \left( r(\mathbf{x}_i) - 1 \right) \nabla_{\boldsymbol{\theta}} \log p_{\boldsymbol{\theta}}(\mathbf{x}_i) = -\sum_{i=1}^{m} \nabla_{\boldsymbol{\theta}} \log p_{\boldsymbol{\theta}}(\mathbf{x}_i), \quad (1)$$

thereby suppressing model likelihood on poor samples. Although the expected gradient remains zero, due to the finiteness of $m$, this now produces nonzero empirical gradients that we can now use for training. (Zhu et al., 2025) shows that incorporating negative RL along with positive examples can be beneficial in LLM training. However, training exclusively on negative examples for an extended period breaks the model's pre-trained knowledge, leading to catastrophic collapse and rendering the model unusable for most tasks. See Fig. 11a in appendix.

## 3.2 SPARSE RL TECHNIQUES: INTRINSIC REWARDS

In response to these well-known challenges, there is a vast literature on RL under sparse rewards. For our purposes, two relevant categories of algorithms can handle all-negative-reward samples in the context of post-training a generative model.

**Count-based methods** Count based methods introduce an exploration bonus based on state visitation counts to reward novelty (Bellemare et al., 2016; Ostrovski et al., 2017). Modern pseudo-count approaches (Ostrovski et al., 2017) employ a neural density model $\rho$ to approximate state visitation. Given an observation $\mathbf{x}$, the density model is updated once to yield a new model $\rho'$, and the intrinsic reward is defined as some increasing function of $\log \rho'(\mathbf{x}) - \log \rho(\mathbf{x})$. Count-based methods do not naturally support conducting multiple updates per reward evaluation; the density model is updated only once (Bellemare et al., 2016; Ostrovski et al., 2017). Applying multiple updates would artificially inflate $\log \rho'(\mathbf{x}) - \log \rho(\mathbf{x})$, producing large bonuses even for non-novel states.

**Random Network Distillation (RND)** RND instead encourages exploration by training two separate networks sharing the same architecture—a *target* network, which is randomly initialized to produce an embedding of an input sample, and a *predictor* network, which is trained to reduce MSE with the predictor network (Burda et al., 2019). The MSE between the target and the predictor is used as a curiosity bonus; when the predictor does not match the target network, it suggests an unfamiliar state, leading to a higher MSE (and exploration bonus). RND can also be used to post-train LLMs (Gao et al., 2025). This method is particularly good for exploring sparse-reward regimes, but like count-based methods, it does not inherently allow for multiple updates per reward evaluation; doing so will decrease the MSE regardless of whether $\mathbf{x}$ is novel or not. This can increase its NREs (Section 5).

## 3.3 REWARD-BASED SAMPLING: GFLOWNET

GFlowNet (Bengio et al., 2021) is designed to sample from a given reward function. Unlike policy gradient and most intrinsic motivation methods, it naturally supports multiple parameter updates per reward evaluation. The most common training objective for GFlowNet is the Trajectory Balance loss $\mathcal{L}_{TB}$ due to Malkin et al. (2022):

$$\mathcal{L}_{TB}(\boldsymbol{\theta}, \hat{Z}) := \frac{1}{m} \sum_{i=1}^{m} \left( \log p_{\boldsymbol{\theta}}(\mathbf{x}_i) - \log \frac{r(\mathbf{x}_i) + \epsilon}{\hat{Z}} \right)^2 = \frac{1}{m} \sum_{i=1}^{m} \left( \log p_{\boldsymbol{\theta}}(\mathbf{x}_i) - \log \frac{\epsilon}{\hat{Z}} \right)^2 \quad (2)$$

where $\hat{Z}$ is a free learnable parameter jointly optimized along with $\boldsymbol{\theta}$, and $\epsilon$ is a small constant to make sure the loss is defined even when $r(\mathbf{x}_i) = 0$. One can fix $\boldsymbol{\theta}$ and solve for $\hat{Z}$ to get the batch-optimal $\hat{Z}$ in a closed form, resulting in the VarGrad-fashion loss function (Richter et al., 2020):

$$\mathcal{L}_{TB_{Vargrad}}(\boldsymbol{\theta}) := \frac{1}{m} \sum_{i=1}^{m} \left( \log p_{\boldsymbol{\theta}}(\mathbf{x}_i) - \frac{1}{m} \sum_{i=1}^{m} \log p_{\boldsymbol{\theta}}(\mathbf{x}_i) \right)^2 . \quad (3)$$

As shown above, the trajectory balance loss becomes the empirical variance of $\log p_{\boldsymbol{\theta}}(\mathbf{x})$ over $m$ samples, so the optimal $p_{\boldsymbol{\theta}}$ assigns an arbitrary constant mass over $m$ samples; the remaining probability mass is distributed uncontrollably. Hence, in the extremely sparse setting, GFlowNet fundamentally cannot learn; the resulting detachment is shown empirically in Figure 11.

## 4 AVOIDING FAILURES WITH BAYESIAN NEGATIVE EVIDENCE LEARNING

We now present BaNEL (Bayesian Negative Evidence Learning). Our aim is to improve the policy's success rate using only reward zero experiences, without any problem-specific surrogate objectives.

**Naive idea.** If our budget for evaluating $r$ were unlimited, we could trivially achieve a perfect success rate by collecting every possible mistake $R := \{\mathbf{x} \in \mathcal{V}^* \mid r(\mathbf{x}) = 0\}$ and avoiding all elements of $R$:

$$p_{\boldsymbol{\theta}|R^C}(\mathbf{x}) \propto p_{\boldsymbol{\theta}}(\mathbf{x})\mathbf{1}[\mathbf{x} \notin R]. \quad (4)$$

Here, $\mathbf{1}[\cdot]$ denotes the indicator function, and we define $p_{\theta|S}(\mathbf{x}) := \frac{p_{\theta}(\mathbf{x})\mathbf{1}[\mathbf{x} \in S]}{\sum_{\mathbf{x}} p_{\theta}(\mathbf{x})\mathbf{1}[\mathbf{x} \in S]}$ given a set $S$. We use $S^C$ to denote the complement in $\mathcal{V}^*$ of a set $S$. This approach is infeasible because the space of failures is combinatorial and we want to minimize NREs. Fortunately, in most tasks, failures exhibit underlying regularities. In such cases, a neural network can learn to recognize and generalize from these patterns, removing the need to encounter every instance. Thus, the key factor determining performance is the model's ability to infer the failure set $R$ from only a limited number of examples. Ideally, we want this ability to scale with compute.

## 4.1 LEARNING A GENERATIVE MODEL FOR FAILED (ZERO-REWARD) ATTEMPTS

We cast the problem of learning regularities in failures as another, in-loop generative modeling problem. Specifically, we train a separate likelihood-based generative model $p_{\phi}$ (parameterized by $\phi$) on $m$ negative examples with the standard maximum likelihood objective:

$$\max_{\phi} \frac{1}{m} \sum_{i=1}^{m} \log p_{\phi}(\mathbf{x}_i).$$

Once well-trained, $p_{\phi}(\mathbf{x})$ can be used to assess whether a given input resembles previously observed failures; specifically, we use $p_{\phi}$ to define a rejection region $\tilde{R}$ approximating $R$.

For that, the rejection region $\tilde{R}$ should contain samples that are likely for $p_{\phi}(\mathbf{x})$ so the model can avoid making similar mistakes to previously-made ones. To this end, we define $\tilde{R}$ as follows:

$$\tilde{R} := \left\{ \mathbf{x} : \frac{p_{\theta}(\mathbf{x})}{p_{\phi}(\mathbf{x})} < \tau \right\} \tag{5}$$

where $\tau$ is a (potentially data-dependent) threshold value. Note that this requires $p_{\theta}$ and $p_{\phi}$ to be likelihood-based generative models under which we can compute the likelihood. Using the rejection region $\tilde{R}$, we form a Bayesian posterior $\tilde{p}_{\theta}$ to approximate $p_{\theta|R^C}$:

$$p_{\theta|\tilde{R}^C}(\mathbf{x}) \propto p_{\theta}(\mathbf{x})\mathbf{1}[\mathbf{x} \notin \tilde{R}], \tag{6}$$

This policy filters out data points that are similar to prior failures according to $\tilde{R}$; equivalently, we direct the model to sample only from $\tilde{R}^C$.

**Success rate analysis.** Recall that success rate is defined as $\rho(p) := \sum_{\mathbf{x}} p(\mathbf{x})r(\mathbf{x})$. The success rate of the posterior can be written as follows:

$$\rho(p_{\theta|\tilde{R}^C}) = \sum_{\mathbf{x} \in \tilde{R}^C} p_{\theta|\tilde{R}^C}(\mathbf{x})r(\mathbf{x}) = \sum_{\mathbf{x} \in \tilde{R}^C} \frac{p_{\theta}(\mathbf{x} \in \tilde{R}^C | \mathbf{x})p_{\theta}(\mathbf{x})}{p_{\theta}(\tilde{R}^C)}r(\mathbf{x})$$

$$= \frac{1}{p_{\theta}(\tilde{R}^C)} \sum_{\mathbf{x} \in \tilde{R}^C} p_{\theta}(\mathbf{x})r(\mathbf{x})$$

$$= \frac{1}{1 - p_{\theta}(\tilde{R})} \left( \rho(p_{\theta}) - \sum_{\mathbf{x} \in \tilde{R}} p_{\theta}(\mathbf{x})r(\mathbf{x}) \right)$$

$$= \frac{\rho(p_{\theta})}{1 - p_{\theta}(\tilde{R})} - \frac{p_{\theta}(\tilde{R})}{1 - p_{\theta}(\tilde{R})}\rho(p_{\theta|\tilde{R}}),$$

where we abuse notation to denote $p_{\theta}(S) = \sum_{s \in S} p_{\theta}(s)$ for some set $S$. The above decomposition gives qualitative insights about the desired properties of $\tilde{R}$:

- **Misclassification rate of $\tilde{R}$.** The posterior success rate decreases when $\rho(p_{\theta|\tilde{R}})$ increases, so we need to train $p_{\phi}$ well and define $\tilde{R}$ properly so that $\tilde{R}$ does not misclassify $r = 1$ samples and mistakenly reject them.

- **Make $\tilde{R}$ as large as possible.** If we can drive $\rho(p_{\theta|\tilde{R}})$ close to zero, the posterior success rate is roughly $\frac{1}{1 - p_{\theta}(\tilde{R})}$ times greater than the prior and approaches 1 as $\tilde{R}$ grows.

Nevertheless, $\tilde{R}$ does not need to be perfect, as $\rho(p_{\theta|\tilde{R}}) \leq \rho(p_{\theta}) \implies \rho(p_{\theta|\tilde{R}^C}) \geq \rho(p_{\theta})$.

---

**Algorithm 1** Sequential Filtering (No Distillation)

---

1: **Initialize** iterations $n$.
2: Sample $\{\mathbf{x}_j\}_{j=1}^m \sim p_{\boldsymbol{\theta}}$.
3: **Fit failure model** $p_{\boldsymbol{\phi}^0}(\mathbf{x})$ by maximizing $\frac{1}{m}\sum_{j=1}^m \log p_{\boldsymbol{\phi}^0}(\mathbf{x}_j)$.
4: **for** $i = 1$ **to** $n - 1$ **do**
5:     Sample $\{\mathbf{x}_j\}_{j=1}^m$ from $p_{\boldsymbol{\theta}}(\mathbf{x}) \prod_{k=0}^{i-1} \mathbf{1}\left[\frac{p_{\boldsymbol{\theta}}(\mathbf{x})}{p_{\boldsymbol{\phi}^k}(\mathbf{x})} \geq \tau\right]$
6:     Evaluate $\{r(\mathbf{x}_j)\}_{j=1}^m$. Terminate if $r(\mathbf{x}_j) = 1$ for any $j$.
7:     **Fit failure model** $p_{\boldsymbol{\phi}^i}(\mathbf{x})$ by maximizing $\frac{1}{m}\sum_{j=1}^m \log p_{\boldsymbol{\phi}^i}(\mathbf{x}_j)$.
8: **end for**
9: **return** $p_{\boldsymbol{\theta}}(\mathbf{x}) \prod_{k=0}^{n-1} \mathbf{1}\left[\frac{p_{\boldsymbol{\theta}}(\mathbf{x})}{p_{\boldsymbol{\phi}^k}(\mathbf{x})} \geq \tau\right]$.

---

**Adaptive selection of rejection region** $\tilde{R}$    As the rejection threshold $\tau$ increases, so does $p_{\boldsymbol{\theta}}(R)$, and hence $\tilde{R}$ rejects samples more aggressively. However, the same threshold $\tau$ could result in drastically different rejection regions $\tilde{R}$ for different negative-sample models $p_{\boldsymbol{\phi}}$. To simplify design, we adaptively choose $\tau$ so that we accept a fixed number of $m$ samples in each batch. To generate $m$ samples, we first draw $mf$ samples from the prior, for some filtering factor $f > 1$. We then sort the $mf$ samples in descending order of likelihood ratio $\frac{p_{\boldsymbol{\theta}}(\mathbf{x})}{p_{\boldsymbol{\phi}}(\mathbf{x})}$, and only accept the first $m$ samples. $f = 1$ means $\tilde{R}$ is empty, whereas a larger $f$ indicates that only samples that are much more likely in our prior $p_{\boldsymbol{\theta}}$ than in our negative model $p_{\boldsymbol{\phi}}$ are accepted.

**Relationship with Cross Entropy Method (CEM).** When $\tau$ is chosen adaptively so that exactly $m$ of the $mf$ candidates are accepted, the procedure coincides with the elite-selection step of the cross-entropy method (CEM) (De Boer et al., 2005). The key difference is that CEM ranks candidates by reward, whereas in our setting reward is always zero, so we instead use the likelihood ratio $\frac{p_{\boldsymbol{\theta}}(\mathbf{x})}{p_{\boldsymbol{\phi}}(\mathbf{x})}$ as a surrogate ranker. As a soft alternative, we also tried importance resampling with weights proportional to this likelihood ratio (analogous to replacing CEM's hard cut with weights), but it did not yield consistent improvements. For simplicity, we therefore adopt the CEM-style hard cut.

## 4.2 COMBINING MULTIPLE FILTERS EFFICIENTLY VIA DISTILLATION

The proposal distribution can be refined online by repeating Bayesian updates as new samples arrive. In this sequential approach, rejection regions from earlier rounds can be accumulated by taking their union (i.e., $\tilde{R} \leftarrow \tilde{R} \cup \tilde{R}_{\text{new}}$ where $R_{\text{new}}$ is the new rejection region). This yields Algorithm 1.[1]

However, this algorithm is not practical because of two reasons: (1) it requires maintaining multiple negative models for filtering, and (2) since the prior rarely generates samples outside all the rejection regions, rejection sampling can become very inefficient. We handle this issue by distilling the filtered distribution into the model at each stage, leading to Algorithm 2 (main difference highlighted in blue). Algorithm 2 is theoretically equivalent to Algorithm 1, while being significantly more efficient in practice. In practice, we implement the distillation step via maximum likelihood training, reusing the same $m$ samples to train the failure model for efficiency. This is the approach adopted in our experiments. See Fig. 1 for a visual illustration of the algorithm.

## 5 EXPERIMENTS

We evaluate BaNEL by constructing new sequential generation tasks with extremely sparse rewards. In Sec. 5.1, we evaluate on MNIST (LeCun et al., 1998), where we can visualize exploration. In Sec. 5.3, we test on a challenging subset of GSM8K (Cobbe et al., 2021) reasoning tasks where pretrained models fail. In these experiments, we **deliberately filter out reward one samples** to test an algorithm's ability to learn from zero-reward observations only. We compare BaNEL (ours) to the random network distillation (Burda et al., 2019) and pseudo-count based methods (Ostrovski et al., 2017) baselines. In Appendix 5.2 we provide extra results where the attacker generates digit-addition

---

[1]We omit the partition function of the unnormalized distributions to simplify notation from now on.

---

**Algorithm 2** Sequential Filtering with Distillation

1: **Initialize** $p_{\boldsymbol{\theta}^0}(\mathbf{x}) \leftarrow p_{\boldsymbol{\theta}}$; iterations $n$
2: Sample $\{\mathbf{x}_j\}_{j=1}^m \sim p_{\boldsymbol{\theta}^0}$.
3: **Fit failure model** $p_{\boldsymbol{\phi}^0}(\mathbf{x})$ by maximizing $\frac{1}{m}\sum_{j=1}^m \log p_{\boldsymbol{\phi}^0}(\mathbf{x}_j)$.
4: **for** $i = 1$ **to** $n - 1$ **do**
5:     Sample $\{\mathbf{x}_j\}_{j=1}^m \sim p_{\boldsymbol{\theta}^{i-1}}(\mathbf{x})\mathbf{1}\left[\frac{p_{\boldsymbol{\theta}}(\mathbf{x})}{p_{\boldsymbol{\phi}^{i-1}}(\mathbf{x})} \geq \tau\right]$.
6:     Evaluate $\{r(\mathbf{x}_j)\}_{j=1}^m$. Terminate if $r(\mathbf{x}_j) = 1$ for any $j$.
7:     **Fit failure model** $p_{\boldsymbol{\phi}^i}(\mathbf{x})$ by maximizing $\frac{1}{m}\sum_{j=1}^m \log p_{\boldsymbol{\phi}^i}(\mathbf{x}_j)$.
8:     **Distill** the filter into the model: $p_{\boldsymbol{\theta}^i}(\mathbf{x}) \leftarrow p_{\boldsymbol{\theta}^{i-1}}(\mathbf{x})\mathbf{1}\left[\frac{p_{\boldsymbol{\theta}}(\mathbf{x})}{p_{\boldsymbol{\phi}^{i-1}}(\mathbf{x})} \geq \tau\right]$.
9: **end for**
10: $\boldsymbol{\eta} \leftarrow \boldsymbol{\theta}^n$
11: **return** $p_{\boldsymbol{\eta}}$.

---

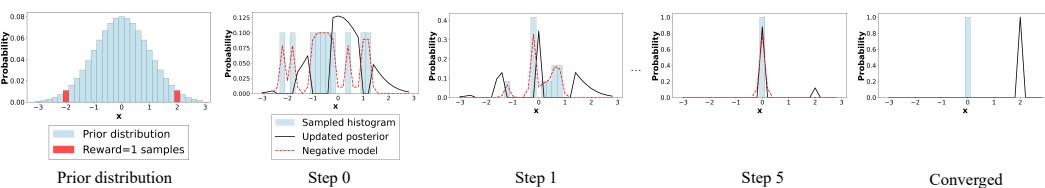

| Prior distribution | Step 0 | Step 1 | Step 5 | Converged |

Figure 1: Illustration of BaNEL on a 1D toy example with negative-reward samples only. The procedure begins with a pre-trained proposal distribution (leftmost). Two reward-one samples (red bars) are located at -2 and 2. At each iteration, the proposal distribution generates samples, which are very likely to be 0-reward. These are used to train a negative model (red dashed curves). The proposal and negative models are combined to form the Bayesian posterior (black curves), following Eq. (6). As iterations progress, the posterior increasingly concentrates on the reward-one regions, until convergence (rightmost).

problems that the target model misanswers. Appendix. B.4 includes ablations that show the effect of various hyperparameters and other design choices regarding the distillation step in Algorithm 2.

## 5.1 MNIST $0 \rightarrow 6$

In this task, we pre-train autoregressive generative models on the 0-digit subset of the MNIST training set, and the task is to discover 6's. Since a 0 is visually close to a 6 digit, pre-training increases the success rate significantly. At the same time, a 6 can only be discovered by doing a significant exploration from 0, testing the algorithm's ability to generate new knowledge.

To summarize our setting: Our pre-trained model $p_{\boldsymbol{\theta}}$ is an autoregressive transformer trained on 0 digits. Our reward $r(\mathbf{x}) = 1$ if the model generates data exactly matching any element of the *target set*, a set of 50,000 6-digits generated by applying random affine transformations to the MNIST 6-digits in the test set. This experimental setting has *extreme reward sparsity*. The base model's success rate is 8e-26 (as $p_{\boldsymbol{\theta}}$ is an autoregressive model, we can evaluate the exact success rate by, e.g., using $\text{torch.logsumexp}()$). We set the total NRE budget to 7500 for all methods.

**BaNEL's success rate scales with compute** Unlike prior sparse RL techniques, BaNEL can utilize additional compute to improve its success rate, even for a fixed number of NREs. Fig. 3 shows that the performance of BaNEL tends to increase as the number of epochs used to train $p_{\boldsymbol{\phi}}$ at each stage increases unlike other two methods. This indicates that while the benefit of BaNEL becomes effective when additional computation is available to extract richer knowledge from failures (unlike our baselines, which cannot exploit additional computation).

Fig. 2 shows that, in the posterior samples, digits shaped like a '0' with the right side removed—thereby resembling a '6'—occur more frequently than in the prior.

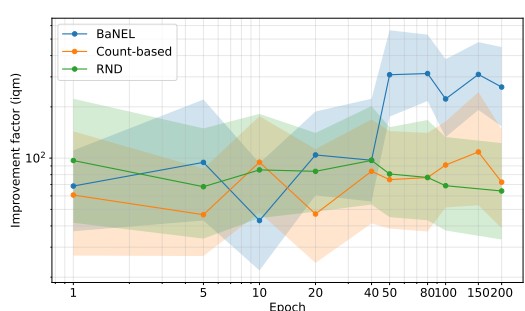

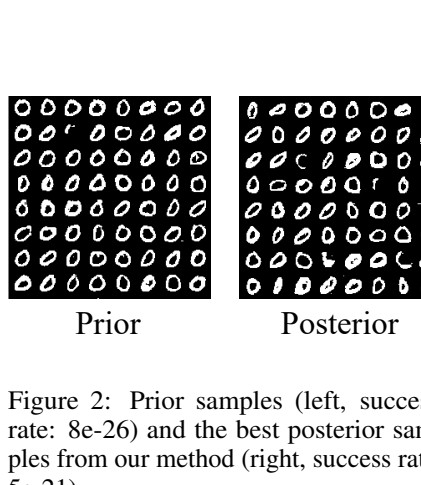

Prior          Posterior

Figure 2: Prior samples (left, success rate: 8e-26) and the best posterior samples from our method (right, success rate: 5e-21).

Figure 3: Compute scaling: Interquartile mean (IQM; mean of the middle 50%) improvement factor in success rate over the base model for BaNEL, count-based, and RND as a function of the number of training epochs. For BaNEL, the x-axis is the number of epochs used to train $p_\phi$ at each stage; for RND and count-based methods, it is the number of epochs used to train the random network and density model per rollout. IQM values are computed over 100 random seeds. Shaded regions indicate 95% bootstrap confidence intervals.

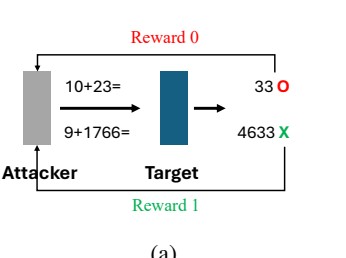

```
# Leading zeros
000840040+6336084=
04967+660843=
006509+602096=

# Carry-chain stressors
4057539400+6460920=
5108069997+50003=
99999999+9=
```

| Pattern | Rate (%) |
|---|---|
| Pre-trained | 0.04 |
| Carry chain | 99.02 |
| Leading zeros | 99.96 |

(a)                          (b)                          (c)

Figure 4: (a) Adversarial attack setup for Sec. 5.2; (b) examples of successful attacks found by BaNEL; (c) rule-based attack results using patterns in (b).

## 5.2 ADVERSARIAL ATTACK ON TOY LANGUAGE MODEL

In this task, the goal is to attack the *target model*, an autoregressive transformer trained to answer digit-addition queries (e.g., it receives 10+23= and must generate 33). The goal of the *attacker model*, also an autoregressive transformer trained to generate questions such as 10+23=, is to propose syntactically valid addition queries on which the target model produces an incorrect sum. Both models use the GPT-2 architecture (we use nanoGPT) with a character-level tokenizer; the vocabulary comprises the ten digits $\{0, \ldots, 9\}$, arithmetic symbols (e.g., +, =), and alphabetic characters. The maximum length of each operand is set to 10. We define the reward as follows:

Table 2: Best interquartile mean (IQM; mean of the middle 50%) improvement factor in success rate over the base model for BaNEL, count-based, and RND. The upper and lower values of 95% bootstrap confidence intervals are also reported. Pre-trained model's success rate is roughly 0.0004. IQM values are computed over 100 random seeds.

| Method | IQM | CI lower | CI upper |
|---|---|---|---|
| Ours | **48.51×** | **34.74×** | **66.93×** |
| Count-based | 1.62× | 1.56× | 1.67× |
| RND | 1.45× | 1.43× | 1.47× |

$$r(\mathbf{x}) = \begin{cases} 1, & \text{if } \mathbf{x} \text{ is a syntactically valid arithmetic expression and the target's output is incorrect,} \\ 0, & \text{otherwise,} \end{cases}$$

and the target is evaluated using greedy decoding. Because grammatically invalid sequences receive zero reward by construction, pre-training the attacker on the same distribution of digit-addition problems is necessary so that it reliably proposes syntactically valid expressions that the target can parse and attempt to answer. See Fig. 4(a) for visual explanation. Since the target is trained well, the pre-trained attacker's empirical success rate is roughly 0.0004 (Clopper-Pearson CI: [0.00032, 0.00047]; num_samples= 300,000, $\alpha = 0.05$).

Table 2 shows that BaNEL outperforms other methods by a large margin. In addition to increasing the raw success rate, this experiment surfaced several qualitative patterns. Fig. 4 (b) shows two examples of successful attacks. BaNEL identifies two failure modes of the target: *(1) Leading zeros*: when at least one of the input digits start with at least one zero, the output result tends to be incorrect. Note that the attacker model had never seen leading zeros during pre-training. *(2) Carry-chain stressors* refer to examples that need to carry a digit during summation. Together, these two failure classes explain a large fraction of successful attacks found by BaNEL.

Based on the insights discovered by BaNEL, we write a script to generate questions following these two patterns to attack the target model. Specifically, we generate 512 samples from each pattern, and compute the resulting success rate. Fig. 4(c) shows that the final success rate is near 1. This suggests that BaNEL can be used both to increase a numeric success rate, but it can also be useful to guide human intuition on hard problems to extract qualitative insights. See Appendix B.1 for more details on how the rule-based attacks are generated. For completeness, we provide additional results where we do not allow for leading zero attacks (Appendix B.2).

## 5.3 GSM8K-HARD

Next, we compare BaNEL with RND (following the implementation of Gao et al. (2025)), the strongest baseline on MNIST setting, on a challenging subset of GSM8K dataset (Cobbe et al., 2021). We select 6 questions from the GSM8K test split on which the Qwen 2.5 0.5B Instruct model (Team et al., 2025), RL fine-tuned with PPO on the same dataset (achieving 0.53 mean@5—average per-problem fraction correct over five attempts—on the test set), attains a success rate between $1 \times 10^{-4}$ and $3 \times 10^{-3}$. This range is small enough to reflect the challenge of sparsity, yet not so small that empirical estimation of success rates becomes impractical. Specifically, we choose the following question IDs: 143, 1248, 1012, 510, 942, and 205. We then further train separate runs, one per selected question. We set the NRE budget to 7680.

As shown in Fig. 5, **BaNEL strictly outperforms RND on 4 problems (143, 205, 1012, and 942)**, achieving higher success rates with significantly fewer NRE. On one problem (1248), BaNEL achieves a comparable success rate while requiring roughly $6\times$ fewer NREs, and on the remaining problem (510), RND outperforms BaNEL. These results demonstrate that BaNEL learns and generalizes more effectively than RND from failure-only feedback. Note that Fig. 5 shows the historical maximum success rate of each baseline. This is an appropriate visualization because the NREs are only an upper bound; in practice, one can always use fewer. The raw values are plotted in Fig. 9.

## 6 DISCUSSION, LIMITATIONS, AND FUTURE WORK

**Limitations** We observe that the success rate of our method does not increase monotonically with training. See Fig. 9 in appendix. Instead, like the RND and count-based method baselines, it peaks at an intermediate stage before declining. We attribute this behavior to two main factors. First, as the generative model shifts toward regions of higher reward, it increasingly produces samples close to high-reward examples, which leads to $\tilde{R}$ containing a greater proportion of incorrect (i.e., reward = 1) samples. Second, errors introduced during the distillation step of the algorithm can accumulate over time. This limitation is not unique to our approach but is shared by all methods that rely on sparse rewards: the success rate cannot be reliably estimated until we discover high-reward samples, making it difficult to determine when training should be stopped. One potential remedy is to design a mechanism that gradually slows the posterior update according to a decaying schedule.

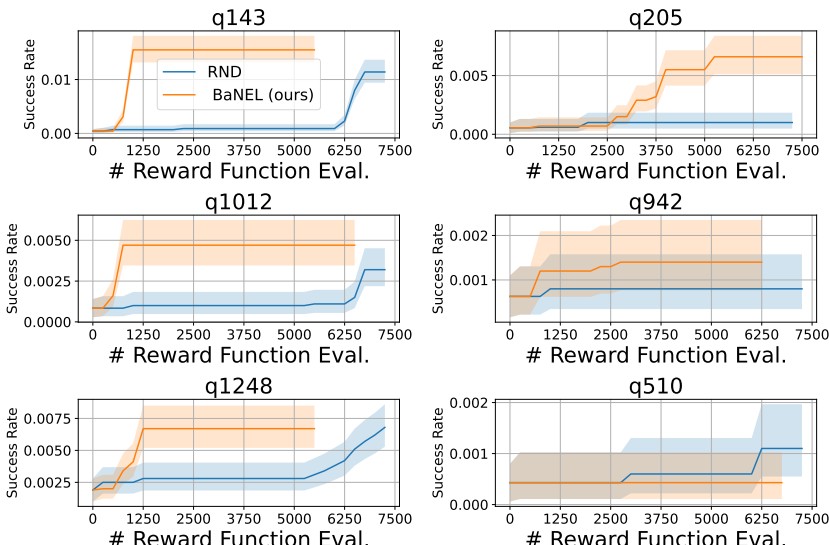

Figure 5: Cumulative best success rate of BaNEL and RND on GSM8K-Hard questions. Shaded area represents confidence intervals (Clopper-Pearson, $\alpha = 0.05$, sample_size=10000).

Such a schedule could be designed using minimal knowledge of a problem such as expected difficulty level.

**Parameterizing $p_\phi$**  Maintaining a separate model $p_\phi$ can be expensive for large models. As an alternative, we explored modeling the negative distribution by conditioning the policy on a negative prompt (e.g., "generate an incorrect answer"). However, we found that training such prompt-conditioned models inadvertently alters the behavior of the original policy, introducing unwanted confounding variables. As such, we avoid sharing the parameter between two models to isolate the effect of applying BaNEL's Bayesian updates. One could leverage low-rank adaptation (LoRA) (Hu et al., 2022) to mitigate this coupling between two models, which we leave to future work.

**Learning fast and slow**  One promising way to tackle the reward sparsity is to execute a learned learning algorithm that adapts from failures and refines its next actions. This can be more flexible and powerful than executing any hand-designed algorithms, including ours. Sequence models such as recurrent neural networks or transformers can serve as *fast learners* (Duan et al., 2016), executing learning algorithms during inference. For instance, transformers can be trained in multi-turn settings, after which they can carry out sophisticated adaptive behavior in context. However, fast learners require a slow learning algorithm to train them. In practice, this means that methods like ours can play a crucial role in providing the outer-loop optimization signal. For instance, applying our algorithm on the level of meta-trajectories to train the parameters of a fast learner is an interesting direction.

## 7 CONCLUSION

We present BaNEL, a method for post-training generative models in extremely sparse reward settings, where models may never encounter positive examples during training. Unlike existing exploration methods such as count-based bonus methods and random network distillation, BaNEL's ability to recognize and generalize from failures scale with compute. Empirical results demonstrate that BaNEL achieves success rates on challenging tasks higher than competitive baselines under the same reward evaluation budget.

## REPRODUCIBILITY STATEMENT

We provide detailed information to facilitate reproducibility of our results, including pseudo-code in Algorithm 2, experiment settings in Sec. 5, and additional implementation details in Appendix B.1. We plan to release our code publicly to further support reproducibility.

## ETHICS STATEMENT

This paper raises ethical concerns similar to other papers on deep generative models. Generative models can produce harmful contents, such as disinformation and violent text. Our experiment on adversarial attacks against a language model (Appendix 5.2) illustrates a potential misuse scenario. However, it is conducted in a controlled, toy setting that does not pose direct risk of harm.

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

# A   ADDITIONAL RELATED WORK

## A.1   HINDSIGHT RELABELING IN RL

One key component of our method is a generative model maximizing the likelihood of failed attempts. Goal-conditioned RL methods such as Andrychowicz et al. (2017); Rauber et al. (2017) use a conceptually similar idea where they train a model conditioned on the suboptimal goal states achieved by the model. Decision Transformer (Chen et al., 2021) and RL upside down (Schmidhuber, 2019) condition the model on scalar reward signals. However, a crucial difference is that we do not merely train a model on failed attempts but use it as a likelihood function to obtain the Bayesian posterior.

## A.2   INTRINSIC REWARDS FOR LANGUAGE MODELS

Beyond the earlier literature focusing mainly on randomly initialized policies, recent works have applied intrinsic rewards such as RND (Gao et al., 2025), entropy bonus (Shen, 2025), or self-consistency (Zhang et al., 2025) to pre-trained LLMs. However, they did not consider extremely sparse settings.

## A.3   BAYESIAN OPTIMIZATION WITH DATA PRIOR

Bayesian Optimization (BO) (Garnett, 2023) shares the goal of maximizing some utility function defined with respect to the reward function while minimizing the number of function evaluations. Although the standard BO formulation does not incorporate the generative prior $p_{\boldsymbol{\theta}}(\mathbf{x})$ (which is different from the function prior used in standard BO) as ours, a few recent works (Hvarfner et al., 2022; Souza et al., 2021) suggest incorporating the data prior into BO.

The belief update in BO relies on *discriminative models* $Pr(r \mid \mathbf{x})$ given observations so far, which is typically modeled as Gaussian Processes or Bayesian Neural Networks (Garnett, 2023). In contrast, our method uses *generative models* as the likelihood function, so we can use autoregressive transformers, which have been shown to scale extremely well.

## A.4   DATA-DRIVEN BLACK-BOX OPTIMIZATION

Recent works on data-driven black-box optimization (Krishnamoorthy et al., 2023; Li et al., 2024) assume access to a large corpus of unlabeled data together with a small set of reward-labeled samples. The typical goal is to optimize a black-box objective by leveraging these offline datasets. A common approach is to train a reward-conditional generative model and then synthesize high-reward candidates by conditioning on desired reward levels. In contrast, we study the online setting, where the model must interleave acquiring new data and updating itself. Moreover, we focus on an extreme regime of sparsity, where the data contain no positive-reward examples, so a reward-conditioned model cannot be meaningfully conditioned on unseen positive reward values. Lin et al. (2022) trains conditional GANs in an online setting, where a classifier is trained on labeled data and its confidence scores are used to guide exploration. However, in the regime where all observed rewards are zero, the classifier cannot be trained meaningfully, and thus its confidence scores provide no useful guidance.

# B   ADDITIONAL EXPERIMENTS

## B.1   IMPLEMENTATION DETAILS

In this section, we provide the detailed settings used in Sec. 5. The distillation step of Algorithm 2 is carried out using maximum likelihood estimation over $m$ samples, with $m$ is 250 for MNIST, adversarial attack experiments, and 256 for GSM8K. We set NRE budget to 30 rounds of exploration, which is equivalent to 7500 and 7680 for MNIST and GSM8K, respectively. Since the sample size is typically insufficient to fully capture the support of the target distribution, the learned model can collapse to a limited subset of modes. To mitigate this issue, at the beginning of each round of BaNEL, we reset the generator's parameters to those of the base model before conducting distillation step, thereby preserving mode coverage.

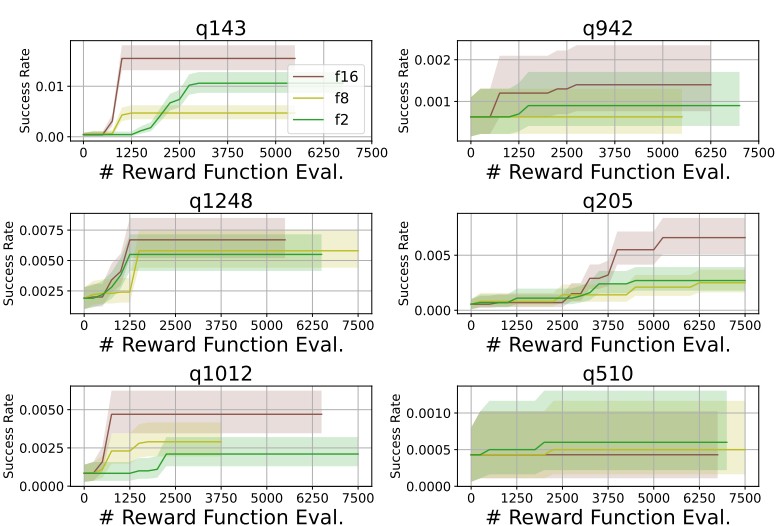

Figure 6: Cumulative best success rate across different filter factors $f$.

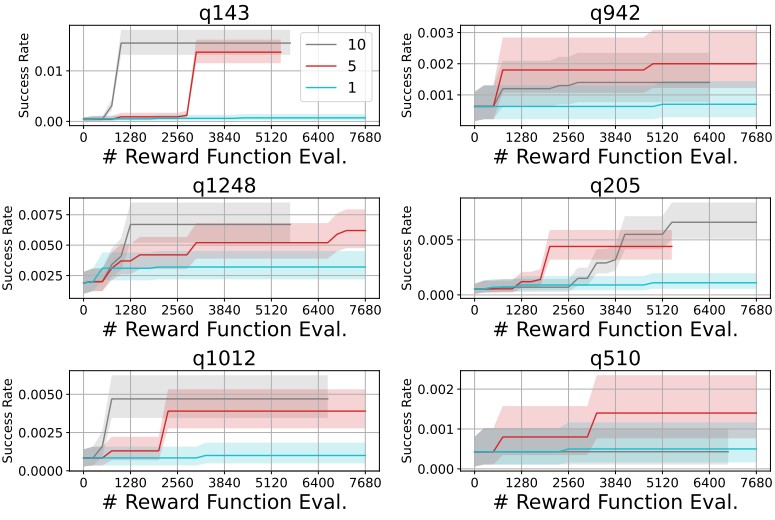

Figure 7: Cumulative best success rate for different numbers of training epochs for $p_{\theta}$.

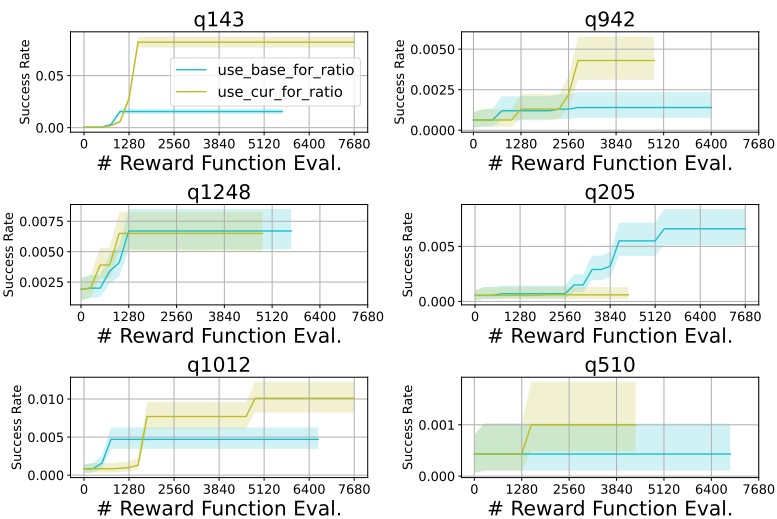

Figure 8: Cumulative best success rate when using the base model $p_{\boldsymbol{\theta}}$ versus the updated model $p_{\boldsymbol{\theta}}^{(i=1)}$ for the likelihood ratio.

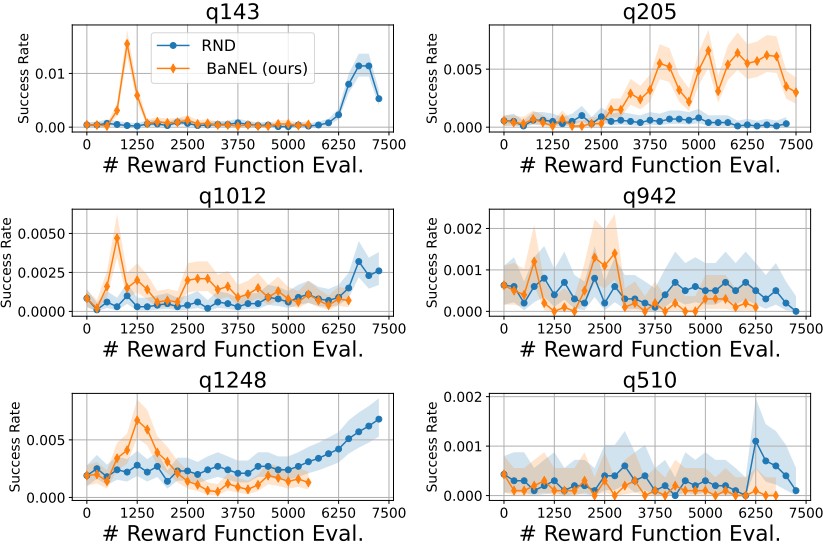

Figure 9: Success rate of BaNEL and RND on GSM8K-Hard questions. Results correspond to Fig. 5.

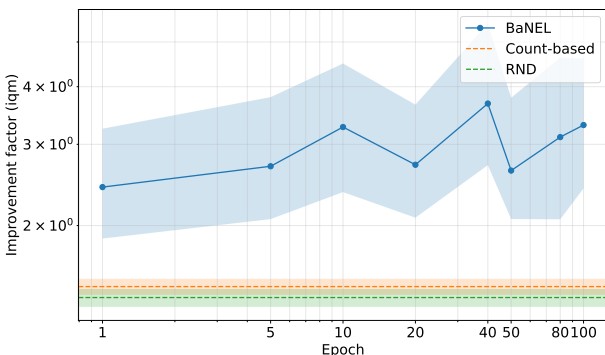

Figure 10: Results on the adversarial attack scenario: IQM improvement factor in success rate over the base model of BaNEL as a function of the number of epochs to train $p_\phi$. Results of Count-based and RND are provided in horizontal lines. IQM values are computed over 100 random seeds. Shaded regions indicate 95% bootstrap confidence intervals. Leading zeros are not allowed (Sec. B.2).

On MNIST, to obtain our best result, $p_\theta$ and $p_\phi$ are trained for 15 and 150 epochs per round, respectively. For adversarial attack experiments, $p_\theta$ and $p_\phi$ are trained for 10 and 100 epochs per round, respectively. On GSM8K, $p_\theta$ and $p_\phi$ are trained for 10 and 5 epochs per round, respectively. The filter factor $f$ is set to $f = 2$ for MNIST, $f = 1.032$ for adversarial attack, and $f = 16$ for GSM8K-hard.

When data have variable lengths, computing $\frac{p_\theta(\mathbf{x})}{p_{\phi^k}(\mathbf{x})}$ and ranking samples within a batch can introduce length bias. To mitigate this, in practice we normalize log-likelihoods by length and compute $\frac{p_\theta(\mathbf{x})^{1/l(\mathbf{x})}}{p_{\phi^k}(\mathbf{x})^{1/l(\mathbf{x})}}$, where $l(\mathbf{x})$ is the length of $\mathbf{x}$. For Qwen 0.5B model, we use the maximum response length of 512.

**Baselines.** For the count-based baseline, we use the same architecture for $p_\theta$ and the density model $\rho$, both initialized with the same pre-trained weights. We adopt the same decay schedule and exploration bonus as in Ostrovski et al. (2017). To improve performance, we additionally apply KL regularization between the current and initial policy. We find that a coefficient of 0.05 works the best for both MNIST and adversarial attack experiments. For the RND baseline on MNIST, we follow the setup of Burda et al. (2019), with the modification that larger models for both the predictor and target yield better performance. Specifically, we use a 4-layer fully connected network with hidden dimension 1024. We regularize with a KL penalty of strength 0.01. For the adversarial attack and GSM8K, we adopt the implementation of Gao et al. (2025). We find that training does not improve success rates without KL regularization. For the adversarial attack experiment, we find that a penalty coefficient of 0.5 works the best for the experiments in Sec. 5.2. For Sec. B.2, 0.01 works the best. For GSM8K, we find that a penalty coefficient of 0.05 works well.

**Rule-based attack for Sec. 5.2.** For carry chain attack, we generate 10-digit addition problems by first sampling the least significant digit pair whose sum is at least 10 to initiate a carry. The remaining digit pairs are sampled to sum exactly to 9 (except for the most significant digit), which propagates the carry when combined with the incoming carry-in of 1. For leading zero attack, we prepend leading zeros with random length to one or both operands of randomly generated addition problems while respecting the 10-digit length constraint.

### B.2 ADDITIONAL RESULTS FOR ADVERSARIAL ATTACK EXPERIMENT

Leading zeros are one of two failure modes of the target model discovered by BaNEL in Sec. 5.2. To ensure that BaNEL's performance gain is not simply due to its ability to discover leading zeros, here we modify the definition of $r$ such that it gives 0 for strings with leading zeros (i.e., leading zeros are now syntactically invalid). Fig. 10 shows the compute scaling result for this setup. Similarly to Fig. 3, BaNEL consistently outperforms other two baselines regardless of the number of epochs used.

Table 3: Wall-clock runtime (in seconds) for BaNEL, pseudo-count, and RND in the MNIST experiment.

| Ours | Count-based | RND |
|------|-------------|-----|
| 952  | 395         | 393 |

### B.3 RUNTIME COMPARISON ON MNIST

In Table 3, we compare the runtime of RND, count-based, and BaNEL with a single NVIDIA H100 GPU. BaNEL uses 150 epochs for $p_\phi$, which incurs additional cost.

### B.4 ABLATION STUDIES FOR GSM8K-HARD

This section presents experiments for some important design choices of BaNEL.

**Filter factor $f$**  Fig. 6 shows the effect of the filter factor $f$. We find that $f = 16$ performs best on this dataset, although all values improve the success rate over the base model for most questions.

**Number of epochs**  In Fig. 7, we sweep over values $1, 5, 10$ for the number of epochs when training $p_\theta$ at each round, and observe that 10 yields the strongest results.

**Computing likelihood ratio with the current proposal**  Algorithm. 2 requires maintaining three models: the current generator $p_{\theta^{i-1}}$, the negative model $p_{\phi^{i-1}}$, and the base model $p_\theta$, which can be computationally costly. However, notice that $p_{\theta^{i-1}}(\mathbf{x}) \propto p_\theta(\mathbf{x})$ for $\mathbf{x} \in \mathrm{supp}(p_{\theta^{i-1}})$ if the distillation is performed optimally. Hence, we can use $\frac{p_{\theta^{i-1}}(\mathbf{x})}{p_{\phi^{i-1}}(\mathbf{x})}$ instead of $\frac{p_\theta(\mathbf{x})}{p_{\phi^{i-1}}(\mathbf{x})}$ to rank samples, as this does not change the relative ordering. Doing so eliminates the need to store the base model, reducing space complexity. As shown in Figure 8, the results are mixed. We use the base model for the likelihood ratio in Sec. 5.

**High-temperature sampling**  A straightforward way to encourage exploration is to increase the sampling temperature. We tested this by applying temperatures of 1.1 and 1.2 to the base model on question 942. While this substantially increased the joint entropy, the resulting success rates were only 0.0005 and 0.0006, respectively, based on 10,000 samples. For comparison, the base model's success rate confidence interval (Clopper–Pearson, $\alpha = 0.05$, $n = 10,000$) is $[0.00016, 0.0011]$. Thus, higher temperatures did not yield a statistically significant improvement. This suggests that reward sparsity cannot be overcome simply by injecting randomness through higher temperature; instead, systematic elimination of failed attempts is required.

**Success rate trends**  Fig. 9 shows that the success rates of BaNEL often peak and then decline. RND exhibits similar behavior for problems 143, 1012, and 510. For the remaining problems, RND either fails to improve the success rate at all or exhausts the NRE budget before reaching its peak.

### B.5 COMPARISON OF NEGATIVE-RL, GFLOWNET, AND BANEL

Figure 11 presents the training dynamics of Negative-RL, GFlowNet, and BaNEL. Starting from a prior model pretrained on MNIST 0-digits, we observe that training of both Negative-RL and GFlowNet collapses, indicating that these methods are not suitable in our extremely sparse reward setting.

## C THE USE OF LARGE LANGUAGE MODELS

LLMs were employed to improve the clarity of several sentences.

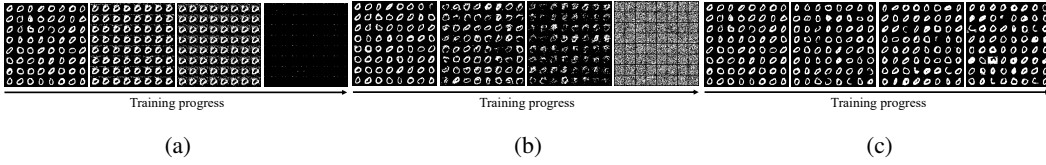

(a)             (b)             (c)

Figure 11: Results of post-training an autoregressive transformer trained on MNIST 0-digits: (a) Negative RL (Eq. (1)); (b) GFlowNets (Eq. (3)); (c) BaNEL (Ours). Both negative RL and GFlowNets result in severe detachment from $p_{\boldsymbol{\theta}}$, rendering the model unusable for most tasks.

