# OpenReview forum: "BaNEL: Exploration Posteriors for Generative Modeling Using Only Negative Rewards"
_ICLR.cc/2026/Conference — Submitted to ICLR 2026_

### Official Review · Reviewer_nZNz · 2025-10-28

**Soundness:** 3
**Presentation:** 3
**Contribution:** 3
**Rating:** 6
**Confidence:** 3

**Summary:**

This paper proposes Bayesian Negative Evidence Learning (BaNEL), which is an algorithm for post-training generative models using only negative rewards (failed attempts). BaNEL minimizes the number of reward evaluations (NREs) and learns the structure underlying failures. The proposed approach learns efficiently from a small number of failures, by allowing for multiple parameter updates for each collected experience. BaNEL uses Bayesian updates of the prior, which does not decrease the model’s likelihood for failed attempts, and thus better preserves the model’s pre-trained knowledge. The authors provide an experimental evaluation that shows BaNEL can achieve a success rate that is several orders of magnitude higher than other competing approaches for the same NRE budget.

**Strengths:**

* The paper provides a good overview of existing algorithms for post-training generative models from reward functions, including policy gradient, sparse RL techniques, and GFlowNet.
* The contributions appear to be novel and technically sound.
* The experimental results are reasonably thorough and show that compared to competing approaches, BaNEL indeed achieves substantial improvements in success rates for a fixed NRE budget. The experiments also show that BaNEL can improve its success rate using additional compute.

**Weaknesses:**

* The primary evaluation metric in the experimental results is success rate, with only a brief qualitative example of the quality and diversity of generated samples, as shown in Figure 2. A more extensive evaluation in terms of the quality and diversity of generated samples should be provided.
* No results on the computational runtime complexity are provided. How much more computationally expensive is BaNEL compared to existing methods for post-training generative models from reward functions?
* BaNEL systematically narrows the search space. However, exclusively learning from negative samples could lead the model to avoid entire regions of the solution space that might contain the optimal solution, because initial explorations in that area failed. This could result in the model converging to a suboptimal solution. Is this an issue in practice? If so, an analysis of this issue should be included in the paper.

**Questions:**

The authors should provide answers to the questions/issues listed under “weaknesses” above.

---

> ### Author Response · Authors · 2025-12-03
>
> We appreciate your time and effort to provide detailed feedback.
>
> ### The primary evaluation metric in the experimental results is success rate, with only a brief qualitative example of the quality and diversity of generated samples, as shown in Figure 2. A more extensive evaluation in terms of the quality and diversity of generated samples should be provided.
>
> We have added an additional experiment in Sec. 5.2, where we show a more detailed qualitative analysis of the generated samples. In our setup, the quality is measured by success rate; visual quality of generated digits is not our goal, unlike in typical image generative modeling. Regarding diversity, we expect BaNEL to preserve diversity better than RL methods as its Bayesian update preserves mode coverage of the prior outside the rejection region.
>
>
> ### No results on the computational runtime complexity are provided. How much more computationally expensive is BaNEL compared to existing methods for post-training generative models from reward functions?
>
> Thanks for the suggestion. We have added the comparison table to Appendix B.3.
>
>
> ### BaNEL systematically narrows the search space. However, exclusively learning from negative samples could lead the model to avoid entire regions of the solution space that might contain the optimal solution, because initial explorations in that area failed. This could result in the model converging to a suboptimal solution. Is this an issue in practice? If so, an analysis of this issue should be included in the paper.
>
> This explains why the success rate for BaNEL (and other methods) peaks and then declines. We discuss this limitation explicitly in the paper (see the limitation section).

---

### Official Review · Reviewer_skBy · 2025-10-30

**Soundness:** 2
**Presentation:** 2
**Contribution:** 2
**Rating:** 4
**Confidence:** 4

**Summary:**

This paper introduces BaNEL (Bayesian Negative Evidence Learning), a new post-training algorithm for generative models that operates entirely from negative (zero-reward) samples under conditions of extreme reward sparsity and expensive reward evaluations.
Instead of relying on positive reward signals or dense feedback, BaNEL trains an auxiliary generative model on failed attempts to estimate a “failure distribution.” It then forms a Bayesian posterior steering generation away from samples that resemble prior failures.
The method supports multiple updates per reward evaluation and integrates a sequential filtering–distillation scheme to progressively refine the policy while keeping computation efficient.
Experiments on MNIST 0→6 generation and GSM8K-Hard reasoning tasks show that BaNEL improves success rates by several orders of magnitude compared to Random Network Distillation (RND) and count-based exploration, despite using the same limited number of reward evaluations.

**Strengths:**

- The research question is important because, as large language models continue to advance rapidly, many challenging tasks appear and remain unsolved due to sparse or extremely limited reward signals. Understanding how to effectively train models under such sparse-reward conditions is therefore a crucial and timely problem.

**Weaknesses:**

- Some part of experiments need further clarification. For example, in MNIST, how success rate of 8e-26 are calculated. From figure 2, there seems no large difference between prior and posterior, which tend to show 5e-21 is just a variance.
- The experiments are limited, especially on GSM8k hard, only a toy setting on 6 subquestions are tested.  A more realistic setting (i.e. a model trained a difficult training set, say several hundred queries and tested on commonly used benchmarks) are expected to verify the effectiveness

**Questions:**

1. This paper discuss a situation that models are unable to receive reward (i.e. very low percent of trajectories can get correct answer), and use negative samples to learn. I am wondering whether this method can be integrated with the setting that we can receive reward (i.e. make the learning more efficient as we can learn from both correct samples and incorrect samples with additional information like BAYESIAN NEGATIVE EVIDENCE). As realistic setting, people can warm up the model with SFT trajectories or use reward model to somehow bypass the zero reward issue even on very complex tasks.
2. achieving 0.53 mean@5 which means it has around 10 successful trajectories for 100 samples, why the success rate is 1e-4?

---

> ### Author Response · Authors · 2025-12-03
>
> We appreciate your time and effort to provide detailed feedback.
>
> ### in MNIST, how success rate of 8e-26 are calculated.
>
> We can compute $\log p_\theta(x)$ explicitly as $p_\theta$ is an autoregressive generative model whose likelihood value is tractable. Then the success rate $\sum_x p_\theta(x)$ can be computed by exponentiating and summing up the log likelihood values. torch.logsumexp is one way to do this in a numerically stable way.
>
> ### From figure 2, there seems no large difference between prior and posterior, which tend to show 5e-21 is just a variance.
>
> During the revision, we made several changes to improve the credibility of our experiments. In particular, we have
> - increased the number of seeds from 5 to 100,
> - included 95% bootstrap CI, and
> - report IQM (the mean of the middle 50%) instead of the mean.
>
> We followed the suggestions by the authors of [2]. In the revised figure, we see a clear benefit of BaNEL when num_epoch is high.
>
> Regarding the visual difference between prior and posterior, the posterior samples indeed become very different visually from the prior if we continue exploration, but we have observed that a large visual gap almost always leads to a worse success rate. Fig. 2 shows the posterior samples that achieve the best success rate, which tend to exhibit smaller visual differences.
>
>
> ### The experiments are limited, especially on GSM8k hard, only a toy setting on 6 subquestions are tested. A more realistic setting (i.e. a model trained a difficult training set, say several hundred queries and tested on commonly used benchmarks) are expected to verify the effectiveness
>
> Our zero-reward problem formulation is different from standard LLM reasoning papers, so it requires a different setup. For example, a train/test split is not applicable here, as the goal is not generalization to the unseen tasks but to solve the current sparse-reward problem. We avoided training the model on multiple queries simultaneously, as each prompt will affect the other, introducing an unwanted confounding factor.
>
> We plan to conduct experiments with larger models and on more difficult problems in the future. At the same time, we want to stress that although GSM8K is normally considered an easy task, in our zero-reward setup, where the reward-1 signal is unavailable, it becomes very challenging.
>
>
> ### This paper discuss a situation that models are unable to receive reward (i.e. very low percent of trajectories can get correct answer), and use negative samples to learn. I am wondering whether this method can be integrated with the setting that we can receive reward (i.e. make the learning more efficient as we can learn from both correct samples and incorrect samples with additional information like BAYESIAN NEGATIVE EVIDENCE). As realistic setting, people can warm up the model with SFT trajectories or use reward model to somehow bypass the zero reward issue even on very complex tasks.
>
> This is a plausible use case. Whether BaNEL outperforms simpler techniques (e.g., decreasing policy likelihoods for failed attempts [1]) remains an open question.
>
> > As realistic setting, people can warm up the model with SFT trajectories or use reward model to somehow bypass the zero reward issue even on very complex tasks.
>
> This could be done for the commonly used RL setup in LLM reasoning papers. However, in our unique setup, there is no known solution to the current problem, so the SFT dataset cannot exist. Training a reward model is also not possible, as all rewards in the training data are zeros.
>
>
> ### achieving 0.53 mean@5 which means it has around 10 successful trajectories for 100 samples, why the success rate is 1e-4?
>
> 0.53 mean@5 (≈53 successes per 100 attempts) is computed over the full test set, which includes easy problems. We then select six problems (out of 1319) on which the model’s success rate is extremely low.
>
>
> ## References
>
> [1] Zhu, Xinyu, et al. "The surprising effectiveness of negative reinforcement in LLM reasoning." arXiv preprint arXiv:2506.01347 (2025).
>
> [2] Agarwal, Rishabh, et al. "Deep reinforcement learning at the edge of the statistical precipice." Advances in neural information processing systems 34 (2021): 29304-29320.

---

### Official Review · Reviewer_tnAv · 2025-11-12

**Soundness:** 1
**Presentation:** 3
**Contribution:** 2
**Rating:** 2
**Confidence:** 4

**Summary:**

This paper presents Bayesian Negative Evidence Learning (BaNEL), an algorithm designed to enhance the query efficiency of post-training where positive reward signals are extremely sparse and reward query is costly. In essence, BaNEL trains an auxiliary density model to fit the distribution of past failed behaviors, and subsequently leverages the density predicted by this model as the criterion for rejection sampling. Specifically, BaNEL filters out samples whose density exceeds some threshold and does not query the reward oracle for these samples, thereby reducing the volume of unnecessary queries to the reward oracle. The performance of BaNEL was empirically validated on a task specifically constructed using data from MNIST and GSM8K. The proposed method demonstrated favorable results in comparison to several established baseline algorithms, including RND and count-based methods.

**Strengths:**

The development of the idea is clear and easy to follow.

The problem addressed by this paper is important. In scenarios where varifiable rewards are not available or obtaining rewards are costly, we need a mechanism to filter samples with high potential for information gain.

**Weaknesses:**

1. The proposed methodology remains a **passive** strategy. While it effectively reduces the number of reward queries by filtering out samples with a high affinity to past failures, it offers no inherent benefit in terms of true sample efficiency. This is because the underlying generative model continues to perform uninformed exploration across the entire sample space. Ideally, an active learning approach should be developed that explicitly models the distribution of failed attempts and actively guides the model away from failure regions.

2. Several claims made within the paper appear **unfounded and conceptually ambiguous**. For example, the authors mentioned that an ideal method should be able to scale with compute (line 51). If this implies that the ability to identify similarities should improve with increased computational resources, it's unclear why methods like RND and count-based exploration are cited as failing to scale with compute (lines 172 and 182). One could easily argue that the RND network, for example, can be updated with multiple epochs to achieve similar scaling. (In fact, I would say RND and BaNEL are conceptually similar in the sense that they are all training some scoring function to identify seen failures, except that BaNEL is using the density from some generative model.)

3. The empirical evaluations appear **insufficiently justified**, making the advantage of the proposed method difficult to comprehend. For example, the success rates in Section 5.1 are truly low, such that I am skeptical whether the benefit exists or not. Figure 3 only presents the averaged results over five seeds without confidence intervals or error bars, which makes it impossible for readers to examine the statistical significance of the scaling trend. Furthermore, the scaling trend appears noisy, and the x-axis pacing is uneven. In Figure 10, it seems BaNEL’s success rate always peaks during early training and then decreases. This is counterintuitive to me, since as the training progresses, the fitted density model should have seen more failure cases and therefore the success rate should instead be higher. Collectively, the presented evidence is insufficient to demonstrate the efficacy of BaNEL.

**Questions:**

1. Is there a specific rationale behind using the ratio p_\theta(x) / p_\phi(x)^\kappa, instead of simply 1 / p_\phi(x)^\kappa? The factor p_\theta(x) seems to downweight the rare samples from the current model.

2. line 370: How did you calculate the exact success rate of generating the digit 6 in Section 5.1? Also what does it mean by using torch.logsumexp to evaluate the exact success rate?

3. Given that the primary objective is to reduce the number of reward queries, a crucial missing baseline is the comparison against methods that train a proxy model or utilize a judge model for reward evaluation. What specific advantages does BaNEL offer over these well-established proxy-model approaches?

---

> ### Author Response · Authors · 2025-12-03
>
> We appreciate your time and effort in providing detailed feedback.
>
>
> ### The proposed methodology remains a passive strategy … the underlying generative model continues to perform uninformed exploration across the entire sample space. Ideally, an active learning approach should be developed that explicitly models the distribution of failed attempts and actively guides the model away from failure regions.
>
> What you describe corresponds to Algorithm 1. In practice, we use Algorithm 2. Here, the underlying generative model does not perform uninformed exploration; it is updated at each stage using Line 7 in Algorithm 2 to avoid sampling previous failures.
>
>
>
> ### Figure 3 only presents the averaged results over five seeds without confidence intervals or error bars, which makes it impossible for readers to examine the statistical significance of the scaling trend.
>
> Thank you for the insightful comment. Following your advice, we made several changes to improve the credibility of our experiments. In particular, we have
> - increased the number of seeds from 5 to 100,
> - included 95% bootstrap CI, and
> - report IQM (the mean of the middle 50%) instead of the mean.
>
> We followed the suggestions by the authors of [5]. We see that when num_epoch is 40 or higher, BaNEL’s benefit becomes clear.
>
> >Furthermore, the scaling trend appears noisy, and the x-axis pacing is uneven.
>
> It is common that the performance of generative models (or any neural networks) does not strictly monotonically increase with epochs; it is a hyperparameter. Still, Fig.3 shows that using sufficient num_epochs (>=40) dominates using an insufficient number of epochs. In Fig.11 of the revised paper, we see a similar trend for the adversarial attack experiment we added.
>
> Regarding x-axis pacing, we tried to add a sufficient number of data points to see the scaling trend, and we did not cherry-pick any of these x-axis values. In Fig. 3, we used two exponentially increasing sequences 5->10->20->40->80 and 50->100->200 in addition to 1 and 150. Note that each data point requires 100 training runs.
>
> ### it's unclear why methods like RND and count-based exploration are cited as failing to scale with compute (lines 172 and 182). One could easily argue that the RND network, for example, can be updated with multiple epochs to achieve similar scaling. (In fact, I would say RND and BaNEL are conceptually similar in the sense that they are all training some scoring function to identify seen failures, except that BaNEL is using the density from some generative model.)
>
> In RND, increasing the number of training epochs simply makes the student network better predict the random-initialized teacher. It is unclear why this should lead to a better exploration bonus; for instance, the prediction loss can collapse universally [2,3]. In fact, some works even recommend the opposite:to slow adaptation by e.g. reducing the predictor learning rate [3] or making the target dynamic [2].
>
> For pseudo-count bonuses, a density model is trained *after* observing a state x to produce $\rho’(x)$, and the bonus is based on $\log \rho′(x) − \log \rho(x)$. If $\rho′$ is trained for many epochs, this gap becomes large regardless of whether $x$ is novel, again making it unclear why this should help.
>
> Fig. 3 in the revised paper shows that neither RND nor the count-based bonus benefits reliably from more epochs.
>
>
>
> ### The success rates in Section 5.1 are truly low, such that I am skeptical whether the benefit exists or not.
>
> After improving our evaluation protocol as described above, we are quite confident that the relative improvements we observe on MNIST are not noise (200X to 600X according to Fig. 3). However, as you noted, the average success rate of the posterior can still be as low as 5e-23. One might ask: even if the relative improvement is real, is the absolute magnitude of improvement meaningful? As an analogy, imagine we increased the likelihood of sampling a cancer cure from GPT-5 by 600X. One could say the improvement is not significant, as the absolute value might still be vanishingly small. On the other hand, we do not believe the research community will fully solve such difficult problems in one paper; this extremely challenging problem setting will most likely require multiple new ideas from multiple papers and research groups.

---

> > ### Author Response · Authors · 2025-12-03
> >
> > ### In Figure 10, it seems BaNEL’s success rate always peaks during early training and then decreases. This is counterintuitive to me, since as the training progresses, the fitted density model should have seen more failure cases and therefore the success rate should instead be higher.
> >
> > If a high-reward sample is similar to previously observed failures, it may fall inside the rejection region, decreasing the success rate. Any method that generalizes from failed attempts may occasionally reject high-reward samples for this reason. As noted in the appendix (B.3 in the revised paper), RND shows similar behavior for problems 143, 1012, and 510. For other problems, RND either fails to improve the success rate or exhausts the NRE budget before reaching its peak. In MNIST/adversarial-attack experiments, we observed that all three methods show the same peak-and-decrease trend.
> >
> >
> >
> > ### Is there a specific rationale behind using the ratio p_\theta(x) / p_\phi(x)^\kappa, instead of simply 1 / p_\phi(x)^\kappa? The factor p_\theta(x) seems to downweight the rare samples from the current model.
> >
> > The rejection region should include samples “likely” under the negative model. Using the raw likelihood value is not ideal for this purpose when $p_\phi$ is a deep generative model [1] (as expected, it did not perform well). A common approach is to use the likelihood ratio if there is a background distribution to contrast against [4]. In our case, we can use $p_\theta$ as a background distribution representing the training data.
> >
> >
> > ### line 370: How did you calculate the exact success rate of generating the digit 6 in Section 5.1? Also what does it mean by using torch.logsumexp to evaluate the exact success rate?
> >
> > We can compute \log p_\theta(x) explicitly as p_\theta is an autoregressive generative model whose likelihood is tractable. Then the success rate \sum_x p_\theta(x) can be computed by exponentiating and summing up the log likelihood values. torch.logsumexp is one way to do this in a numerically stable way.
> >
> > ### Given that the primary objective is to reduce the number of reward queries, a crucial missing baseline is the comparison against methods that train a proxy model or utilize a judge model for reward evaluation. What specific advantages does BaNEL offer over these well-established proxy-model approaches?
> >
> > Our zero-reward problem formulation is different from many LLM reasoning papers and thus makes many existing approaches inapplicable. For example, to train a reward model, one must have at least some high-reward samples; otherwise, the reward model will just predict 0 and not learn anything meaningful. Similarly, one could try using another LLM as a judge, but there is no reason to believe that the LLM judge will provide the correct exploration signal, as it was never trained to do so; again, we do not have any high-reward samples to ground the judge.
> >
> >
> >
> > ## References
> >
> > [1] Nalisnick et al., Do Deep Generative Models Know What They Don't Know?
> >
> > [2] Pecháč, Matej, Michal Chovanec, and Igor Farkaš. "Self-supervised network distillation: An effective approach to exploration in sparse reward environments." Neurocomputing 599 (2024): 128033.
> >
> > [3] Davoodabadi, Mohammadamin, Negin Hashemi Dijujin, and Mahdieh Soleymani Baghshah. "PreND: Enhancing Intrinsic Motivation in Reinforcement Learning through Pre-trained Network Distillation." arXiv preprint arXiv:2410.01745 (2024).
> >
> > [4] Ren, Jie, et al. "Likelihood ratios for out-of-distribution detection." Advances in neural information processing systems 32 (2019).
> >
> > [5] Agarwal, Rishabh, et al. "Deep reinforcement learning at the edge of the statistical precipice." Advances in neural information processing systems 34 (2021): 29304-29320.

---

### Author Response · Authors · 2025-12-03
**General Response**

We notice that reviewers generally agree on the importance of the problem formulation but wanted to see more depth in the experiments. In the revised paper, we therefore add an additional experiment in Sec. 5.2: attacking a digit-addition transformer. To summarize the setup,

- The target model is an autoregressive transformer trained to answer digit-addition queries (e.g., it receives `“10+23=”` and must generate `“33”`).
- The attacker model, also an autoregressive transformer pre-trained on the same dataset to generate queries such as “10 + 23 =”, aims to propose syntactically valid addition problems on which the target model produces incorrect answers.

Since the target is well-trained, the pre-trained attacker’s empirical success rate is approximately 0.0004.
BaNEL improves this success rate by roughly 50X, significantly outperforming the baselines. BaNEL also identifies two key failure modes—leading zeros and carry-chain stressors—this pattern allows us to form rule-based attacks with **near-perfect attack success rates**. Please see the paper for more details.

In addition to this, we made several changes to improve the credibility of our experiments. In particular, we have:
- increased the number of seeds from 5 to 100,
- included 95% bootstrap CI, and
- report IQM (the mean of the middle 50%) instead of the mean,

as suggested by [1].

[1] Agarwal, Rishabh, et al. "Deep reinforcement learning at the edge of the statistical precipice." Advances in neural information processing systems 34 (2021): 29304-29320.

---

### Meta-Review · Area_Chair_Tmte · 2026-01-04

**Summary:**

The paper proposes BaNEL (Bayesian Negative Evidence Learning), a framework for post-training generative models using exclusively negative (zero-reward) samples. The method trains a failure model to define a rejection region and distills this into the policy. The authors demonstrate "compute scaling" properties where training the failure model longer improves performance on MNIST and a toy arithmetic attack task.

While the problem setting (learning from failure) is novel and the authors have improved statistical rigor (100 seeds), the AC recommends rejection due to the following outstanding concerns as pointed out by most reviewers that have not been properly addressed during the rebuttal stage, including (1) insufficient empirical validation: the evaluation on the primary reasoning benchmark (GSM8K) is restricted to a microscopic subset of 6 questions. Drawing generalizable conclusions about the method's efficacy on LLMs from such a limited set is not scientifically sound. The other experiments (MNIST 0$\to$6, Toy Arithmetic) are too simple to demonstrate real-world utility for modern generative modeling; (2) computational cost: as shown in the added experiments, BaNEL is much slower than baselines like RND. While the paper argues for minimizing reward evaluations, the high computational cost of the inner-loop training and distillation makes the trade-off questionable; and (3) training instability.

Overall, the paper presents an interesting direction for learning from negative samples, but the instability/high cost of the algorithm and the limited scope of the experiments fall short of the bar for acceptance.

**Reviewer Concerns:**

Addressed Concerns:
1. Statistical significance: The authors addressed concerns about result variance by reporting Interquartile Means (IQM) over 100 random seeds, which resolved initial worries about the results being noise.
2. Clarification of "vanishing" probabilities: The authors clarified that the extremely low success rates (e.g., $8e^{-26}$) are calculated via torch.logsumexp on autoregressive likelihoods, not empirical count, which resolves the confusion regarding how such small probabilities were measured.
3. Missing runtime & qualitative analysis: The authors added experiments comparing wall-clock runtime, showing BaNEL is slower than RND (but functional), and also provided qualitative examples of the "adversarial attacks" found for interpretability.

Outstanding Concerns:
1. Experimental scope: This remains the most critical concern. Section 5.3 states the evaluation on GSM8K was restricted to only 6 specific questions. While the authors justify this by selecting "hard" instances, evaluating a general-purpose generative method on a sample size of $N=6$ is statistically insufficient for a top-tier conference and struggles to prove robustness.
2. Computational overhead: Despite the compute scaling argument, the results reveal BaNEL is much slower than RND, and can be unstable. Combined with the need to train a separate failure model $p_\phi$ and perform distillation, the computational cost per reward evaluation is high, which may offset the benefit of reducing the number of reward evaluations.

**Reviewer Scores:**

- Reviewer tnAv: remain negative (2)
- Reviewer skBy: remain negative (4)
- Reviewer nZNz: remain positive (6)

---

### Decision · Program_Chairs · 2026-01-26

Reject